# Acute effects of nitrate and breakfast on working memory, cerebral blood flow, arterial stiffness, and psychological factors in adolescents: Study protocol for a randomised crossover trial

**Callum Regan**[1,2], **Emerald G. Heiland**[1,3]*, **Örjan Ekblom**[1], **Olga Tarassova**[4], **Karin Kjellenberg**[1], **Filip J. Larsen**[4], **Hedda Walltott**[1], **Maria Fernström**[1], **Gisela Nyberg**[1,5], **Maria M. Ekblom**[1,6], **Björg Helgadóttir**[1,7]

1 Department of Physical Activity and Health, The Swedish School of Sport and Health Sciences (GIH), Stockholm, Sweden, 2 Division of Physiotherapy, Department of Neurobiology, Care Sciences and Society, Karolinska Institutet, Huddinge, Sweden, 3 Department of Surgical Sciences, Medical Epidemiology, Uppsala University, Uppsala, Sweden, 4 Department of Physiology, Nutrition, and Biomechanics, The Swedish School of Sport and Health Sciences (GIH), Stockholm, Sweden, 5 Department of Global Public Health, Karolinska Institutet, Solna, Sweden, 6 Department of Neuroscience, Karolinska Institutet, Solna, Sweden, 7 Division of Insurance Medicine, Department of Clinical Neuroscience, Karolinska Institutet, Solna, Sweden

* emerald.heiland@gih.se

**Data Availability Statement:** No datasets were generated or analysed during the current study.

## Abstract

### Background

Inorganic nitrate has been shown to acutely improve working memory in adults, potentially by altering cerebral and peripheral vasculature. However, this remains unknown in adolescents. Furthermore, breakfast is important for overall health and psychological well-being. Therefore, this study will investigate the acute effects of nitrate and breakfast on working memory performance, task-related cerebral blood flow (CBF), arterial stiffness, and psychological outcomes in Swedish adolescents.

### Methods

This randomised crossover trial will recruit at least 43 adolescents (13–15 years old). There will be three experimental breakfast conditions: (1) none, (2) low-nitrate (normal breakfast), and (3) high-nitrate (concentrated beetroot juice with normal breakfast). Working memory (n-back tests), CBF (task-related changes in oxygenated and deoxygenated haemoglobin in the prefrontal cortex), and arterial stiffness (pulse wave velocity and augmentation index) will be measured twice, immediately after breakfast and 130 min later. Measures of psychological factors and salivary nitrate/nitrite will be assessed once before the conditions and at two-time points after the conditions.

Due to ethical restrictions, all relevant data will only be made available, after the study completion, upon reasonable request via https://www.gih.se/forskning/forskningsgrupper-och-projekt/fysisk-aktivitet-hallbarhet-och-hjarnhalsa—e-pabs/om-forskningsgruppen.

**Funding:** This project is supported by The Knowledge Foundation https://www.kks.se/ (20180040; ÖE, GN), and the following companies COOP Sverige, IKEA, Skandia, Skanska, Generation Pep, and Konsumentföreningen Stockholm. Neither the funder nor the partner companies had a role in the design of the study, nor will they in data collection or analysis, interpretation of data, or in the preparation or writing of the manuscript.

**Competing interests:** No authors have competing interests: (CR, EGH, ÖE, OT, KK, FJL, HW, MF, GN, MME, BH)

**Abbreviations:** CBF, cerebral blood flow; oxy-Hb, oxygenated haemoglobin; deoxy-Hb, deoxygenated haemoglobin; NO, nitric oxide; PWV, pulse wave velocity; AIx, augmentation index; fNIRS, functional near-infrared spectroscopy; PANAS, Positive and Negative Affect Scale; VAS, visual analogue scale; CVD, cardiovascular disease.

## Discussion

This study will provide insight into the acute effects of nitrate and breakfast on working memory in adolescents and to what extent any such effects can be explained by changes in CBF. This study will also shed light upon whether oral intake of nitrate may acutely improve arterial stiffness and psychological well-being, in adolescents. Consequently, results will indicate if nitrate intake from beetroot juice or if breakfast itself could acutely improve cognitive, vascular, and psychological health in adolescents, which can affect academic performance and have implications for policies regarding school meals.

## Trial registration

The trial has been prospectively registered on 21/02/2022 at https://doi.org/10.1186/ISRCTN16596056. Trial number: ISRCTN16596056.

## Introduction

Inorganic nitrate can be obtained from diet, through sources such as rocket (rucola), spinach, lettuce, and beetroot [1], with green leafy vegetables contributing to the majority of nitrate consumption [2]. Dietary nitrate is converted into nitrite in the oral cavity, with some of the consumed nitrite being reduced in the stomach to nitric oxide (NO) [3]. However, nitrite is also absorbed into the bloodstream and distributed across the tissues in the body. There, nitrite can be reduced to bioactive NO by several different processes [4]. NO is an important molecule involved in many systemic and cerebral physiological processes. Previous studies have demonstrated its beneficial effects on blood pressure [5–9], endothelial function [8, 9], arterial stiffness [3, 7, 10], and mitochondrial respiratory efficiency in human skeletal muscle [11], whereas inadequate NO has been associated with greater risk for cognitive decline [12, 13]. Despite this knowledge, randomised controlled trials (RCTs) examining the effects of dietary nitrate on cognitive function have been few, with small sample sizes, and only in adult populations, resulting in mixed results [14]. Therefore, more studies of higher quality and in other populations that could benefit from the effects of nitrate, such as adolescents, are needed.

NO regulates many neurobiological processes in the brain and vascular system, such as the regulation of synaptic plasticity, immune response, vascular tone, and neurotransmission, thus promoting cerebral blood flow [15]. A diet rich in nitrate can promote this cascade from nitrite to NO, leading to physiologic effects that may improve prefrontal-dominate cognitive tasks, such as tasks of executive function that are necessary for everyday performance [16]. Executive functions are mental processes that require concentration and attention, with working memory being a core executive function. Working memory involves retaining and manipulating limited information temporarily to be used immediately, such as in problem solving and decision making [17]. A previous meta-analysis of RCTs including adult populations found mixed results on the effect of nitrate on cognitive performance, partly due to the large heterogeneity in types of cognitive tests used and small sample sizes [14]. However, two previous RCTs showed beneficial effects of nitrate on executive function, one including a working memory test [16, 18]. In children and adolescents working memory has been shown to be the strongest contributing component in the association between executive functions and academic performance compared with other executive function components [19]. Thus, working memory is

an important function for performing well academically, and a nitrate-rich diet may enhance this, as seen in older adults [20].

Dietary nitrate and NO have been suggested to enhance neurovascular coupling during cognitive tasks [21]. Neurovascular coupling involves an enhancement in regional cerebral oxygen metabolism coupled to a cognitive task, in order to meet the metabolic demands of neural tissues [22, 23]. As a result, there is an excess of cerebral blood flow (CBF; measured as an increase in oxygenated haemoglobin [oxy-Hb] and a decrease in deoxygenated haemoglobin [deoxy-Hb] concentrations) [24] to the activated regions [22, 23]. Increases in CBF to relevant cerebral regions have been observed during increased cognitive workload [25], where a more pronounced response can occur for more difficult tasks [26]. One study found in older adults that a high intake of nitrate-rich foods increased regional cerebral perfusion in the prefrontal cortex (the part of the brain associated with executive functions and working memory), but not globally [20]. Another intervention study of young healthy adults (mean age 21.3 years), demonstrated that ingestion of dietary nitrate from beetroot juice improved working memory on a revised serial subtraction task compared with the placebo condition [16]. The same intervention study measured changes in CBF (indicated by changes in total haemoglobin) through functional near-infrared spectroscopy (fNIRS) and found nitrate to modulate CBF during the serial subtraction task [16]. However, more studies are needed to disentangle these effects as conflicting results have also been demonstrated on the effects of nitrate on cognitive function and CBF [14]. Thus, a better understanding is needed into NO's potential mechanistic effects on working memory performance and CBF changes particularly in adolescents.

Reduced availability and functionality of NO in the body has been associated with endothelial dysfunction and cardiovascular disease (CVD) [27–29]. Arterial stiffness is an independent predictor of cardiovascular risk [30] and is measured via pulse wave velocity (PWV) and augmentation index (AIx). Two meta-analyses of RCTs, of adults, have reported that intake of inorganic nitrate reduced PWV and AIx [3, 31]. This reduction was observed in one of the meta-analyses specifically from inorganic nitrate from beetroot, suggesting that this dietary source may well have a beneficial effect on certain cardiovascular risk factors [31]. Despite this, all of these studies have been in young and older adults and in mixed health populations. To our knowledge, no studies have investigated the effects of nitrate on vascular health in adolescents, thus the effects of nitrate consumption on arterial stiffness remains to be explored in this population.

More widely studied topics in adolescents are the effects of breakfast consumption on cognitive function, cardiovascular health, and psychological factors. There are no specific dietary recommendations for breakfast composition in Sweden. Still, Swedish dietary guidelines can be applied to breakfast, such as: increasing fruit, vegetable, nut, and seed intakes and opting for wholegrain cereals as well as low-fat dairy options [32]. Breakfast consumption has been suggested to be linked with numerous health benefits in adolescents including cognitive performance and cardiovascular health [33]. Adolphus et al. [33] found that consuming breakfast compared to fasting to have positive and same morning effects on cognitive performance. Cognitive tasks requiring attention, executive functioning, and memory were completed more reliably in those consuming breakfast compared to those omitting breakfast among adolescents [33]. In contrast, irregular breakfast habits have been associated with lower academic success [34]. In regards to cardiovascular health, results from the Healthy Lifestyle in Europe by Nutrition in Adolescence study [35] showed that consuming breakfast regularly was associated with a healthier cardiovascular profile (e.g. lower waist circumference, blood pressure, and total content of high-density lipoproteins). Furthermore, in a sample of 795 adolescents, eating breakfast regularly was found to be inversely associated with multiple CVD risk factors [36]. Adolescents that habitually consume breakfast are more likely to have a higher diet quality [37], eat more fruits and vegetables [38], and have a

decreased risk of being overweight and obese [39]. Despite this, the prevalence of breakfast skipping in a sample of 2929 adolescents across nine European countries was found to be 29% [35]. The same study found that females were more likely to be breakfast skippers than males [35]. Moreover, in a recent cross-sectional study with a sample size of 1139, around 40% of Swedish adolescents skipped breakfast at least one time per week [38]. Therefore, still to this day, this important meal is being skipped by many adolescents. In addition, very few studies have been performed looking at breakfast compositional differences than comparing breakfast to breakfast omission [33]. Therefore, the combination of nitrate-rich foods and breakfast on cognitive performance has not been previously examined. Furthermore, whether the effect of breakfast on cognitive performance is explained by CBF changes is unknown. One study of female university students found that skipping breakfast can have negative physiological effects on CBF [40]. Further study is needed to test whether the same results occur in adolescents and on a working memory-related CBF. This study will be the first to investigate various breakfast compositions on cognitive function, and in relation to cerebral physiological effects.

Breakfast consumption compared to breakfast omission has also been found to have positive effects on mood and psychological factors in adolescents [41, 42]. Cooper et al. found that breakfast consumption led to higher self-reports of energy and satiety as well as lower self-reports on tiredness in adolescents [41]. Additionally, higher self-reports of alertness and contentment have been described in those that consumed breakfast compared with those who skipped breakfast [42]. Changes in one's state of psychological well-being and how one is feeling after breakfast consumption may facilitate attention and motivation in adolescents when performing cognitive tasks, thus influencing cognitive function [33]. Therefore, it is useful to measure mood and other psychological factors when assessing cognitive function.

The acute effects of nitrate on cognitive function, cognitive task-related CBF, vascular health, and psychological factors are believed to not have been studied before in adolescents nor has the acute effects of breakfast on these aforementioned outcomes. Thus, there is a large lack of knowledge and a need to investigate the effects of nitrate and breakfast on these important health outcomes. Consuming nitrate-rich beetroot juice with breakfast is considered to be an achievable, ecologically valid, and relatively low-cost method to increase consumption of dietary nitrate. Therefore, the results from this intervention could help to promote consuming nitrate-rich beetroot juice or other nitrate-rich vegetables and could lead to an increase in overall nitrate-rich vegetable consumption, in this population. Nevertheless, adherence to nitrate-rich beetroot juice or vegetables could be a challenge in the adolescent population. Previous research has found the consumption of fruit and vegetables amongst Swedish adolescents to be below 50% of the recommended intake [43]. This study will indirectly explore and provide insight into the adherence and acceptance of consuming nitrate-rich beetroot juice with breakfast in adolescents.

## Aims and research questions

The primary aim of this study will be to assess the effects of having a high-nitrate breakfast on working memory in 13- to 15-year-old adolescents compared with a regular breakfast (referred to as low-nitrate breakfast) or no breakfast. Secondly, we aim to test the effect of having a high-nitrate breakfast compared with a low-nitrate breakfast or no breakfast on cognitive task-related CBF, arterial stiffness, and psychological factors. Thirdly, we aim to examine the effects of having a regular breakfast (defined as low nitrate breakfast) compared with no breakfast on working memory, task-related CBF, arterial stiffness, and psychological factors. Finally, we want to test whether salivary nitrate/nitrite levels will differ across breakfast conditions. The research questions are divided into two parts.

**Part I–Primary research question.** Does having a high-nitrate breakfast significantly improve performance on working memory tests compared with having a low-nitrate breakfast or having no breakfast in 13- to 15-year-old adolescents?

**Part II–Secondary research questions.** Is there a statistically significant difference in cognitive task-related CBF (oxy-Hb and deoxy-Hb), arterial stiffness, and psychological measures 130 min after a high-nitrate breakfast compared with a low-nitrate or no breakfast?

1. Is there a statistically significant difference in working memory, cognitive task-related CBF, arterial stiffness, and psychological factors 130 min after having low-nitrate breakfast compared with no breakfast?

2. Will salivary nitrate/nitrite levels differ across breakfast conditions?

## Hypotheses

**Part I.**

1. Working memory performance will significantly improve from immediately after the breakfast (time point 1) to 130 min after the breakfast (time point 2) in the high-nitrate breakfast condition.

   a. This improvement will significantly differ from the low-nitrate or the no breakfast conditions, where there will be no change or a reduction in working memory performance.

**Part II.**

2. The CBF parameter of oxy-Hb will significantly increase from time point 1 to time point 2 for all breakfast conditions.

   a. This change will be significantly greater for the high-nitrate breakfast condition compared with the low-nitrate and no breakfast conditions.

3. Deoxy-Hb will decrease for all conditions from time point 1 to time point 2.

   a. The decrease will be significantly greater for the high-nitrate breakfast compared with the low-nitrate and the no breakfast conditions.

4. Arterial stiffness will significantly decrease after the high-nitrate breakfast condition.

   a. The decrease will be significantly greater in the high-nitrate breakfast compared with the low-nitrate and no breakfast conditions.

5. Psychological factors (i.e. mood, alertness, and less sleepiness) will improve from time point 1 to time point 2 for the high-nitrate condition

   a. The improvement in psychological factors will be significantly greater for the high-nitrate breakfast compared with the low-nitrate and no breakfast conditions.

6. Working memory performance will significantly improve in the low-nitrate breakfast condition from time point 1 to time point 2.

   a. This improvement will be significantly different from the no breakfast condition.

7. Oxy-Hb will increase from time point 1 to time point 2 for the low-nitrate breakfast condition.

   a. This increase will be significantly greater for the low-nitrate breakfast compared with the no breakfast condition.

8. Deoxy-Hb will significantly decrease for the low-nitrate breakfast condition.

   a.  This decrease will be significantly different from the no breakfast condition.

9. Arterial stiffness will significantly decrease in the low-nitrate breakfast condition.

   a.  This decrease will be significantly different from the no breakfast condition.

10. Psychological factors (i.e. mood, alertness, and less sleepiness) will improve from time point 1 to time point 2 for the low-nitrate breakfast condition

   a.  This difference will be significantly greater for the low-nitrate breakfast condition compared with the no breakfast condition.

11. Salivary nitrate/nitrite levels will be significantly higher after the high-nitrate breakfast at time point 1 and time point 2 compared to the low-nitrate and no breakfast conditions, and higher for low-nitrate vs. no breakfast.

## Methods

### Study design

This study is a randomised crossover trial with three experimental conditions. The experimental procedures will occur at the Swedish School of Sport and Health Sciences (GIH) in Stockholm, Sweden. The researchers will visit the schools of potential participants to carry out familiarisation visits. Alternatively, the potential participants will visit GIH, if deemed more convenient for the participants. A schedule of enrolment, interventions, and assessments can be found in **Fig 1**.

After completing the familiarisation, the participants will participate in three experimental days consisting of three randomly ordered experimental conditions. A minimum washout period of six days will be in place to reduce any carry-over effects from the previous condition [44, 45]. Most often the participants will come in on the same weekday, three weeks in a row. Two participants will be tested on the same test day, with a 20-minute gap in starting times. There will be a 24 hour monitoring and standardisation of physical activity, sleep, and dietary intake for all participants one day before each test-day. Participants will receive a 500 SEK (~ €47) gift card for their participation.

**Ethical approval and consent to participate.**   This study obtained ethical approval from the Swedish Ethical Review Authority, Stockholm, Sweden (Dnr: 2021-07053-01). The trial registration can be found at: https://doi.org/10.1186/ISRCTN16596056. Written informed consent will be obtained by the participants and guardians before data collection begins. Compensation will be provided to any participant in case they are harmed due to the study. This study is part of larger project called "Physical activity for healthy brain functions in school youth". The results from this project will be directly used in the planning of interventions that will be implemented in schools in the Stockholm area.

### Participants

Schools will be selected from the Stockholm region. Both the participants and guardian/s must sign and provide consent before data collection begins. Inclusion criteria for participants are adolescents in grades 7–9 (approximately from ages 13 to 15 years), attending schools in the Stockholm area, and can understand Swedish. Those that have been diagnosed with diabetes, epilepsy, vascular health conditions/circulatory abnormalities, or have visual/auditory impairments will be excluded. Participants will be informed that they can decide to drop out at any time without explanation.

| | | STUDY PERIOD | | | | | | | |
|---|---|---|---|---|---|---|---|---|---|
| | | Enrolment | Allocation | Post-allocation | | | | | |
| TIMEPOINT (days) | | $-t_1$ | 0 | $t_1$ | $t_2$ | $t_3$ | $t_4$ | $t_5$ | $t_6$ |
| ENROLMENT | Eligibility screen | ● | | | | | | | |
| | Informed consent | ● | | | | | | | |
| | Allocation | | ● | | | | | | |
| INTERVENTIONS (randomised order) | A (no breakfast); OR B (low-nitrate breakfast); OR C (high-nitrate breakfast) | | | | ● | | ● | | ● |
| ASSESSMENTS | | | | | | | | | |
| *Familiarisation visit* | Age; sex; food allergies/tolerances; health survey; head size; nitrate/breakfast survey; usual physical activity | ● | | | | | | | |
| *Pre-test day* | Accelerometer measures; diet, sleep, and physical activity diary | | | ● | | ● | | ● | |
| *Outcome variables* | Working memory; cerebral blood flow; arterial stiffness; psychological factors | | | | ● | | ● | | ● |
| *Other variables* | Salivary nitrate; breakfast | | | | ● | | ● | | ● |
| | Height; weight* | | | | ● | | | | |

*Measured on the day of intervention C.

**Fig 1. Schedule of enrolment, interventions, and assessments for a 3-arm, randomised, crossover study design.**

## Familiarisation visit

The researchers will visit the schools prior to randomisation in order to introduce all the procedures that will be carried out on the test days to the participants and perform any initial measurements. This will include practising the cognitive tests (as to reduce practice effects), showing the fNIRS cap, describing how arterial stiffness is measured, and being familiarised with the questionnaires for measuring psychological factors. Demographic and other data for important measures will also be collected, these will include: age, sex, head measurements (i.e. circumference, nasion to inion, and between pre-auricular points for fNIRS cap size and head registration in the analysis), food allergies or any dietary intolerances, and a health declaration questionnaire.

Additionally, a short questionnaire will be provided for the participants, asking them how often they consume various nitrate-rich foods (pictures of foods will be provided) and how often they eat breakfast. This is to assess the proportion of regular consumers of nitrate-rich foods and the proportion of those that habitually consume breakfast. Schedules for organised sport and physical education classes will also be collected in order to standardise physical activity on pre-test days and to organise consistent times for participants to visit the lab for the data collection.

Post-familiarisation, participants will be paired according to head size (for practicality in fNIRS measuring of two participants on the same day) randomised by the researchers via a computer-generated random order of experimental conditions that is equally assigned to

every participant. The randomisation will be password-protected and will not be assigned to the participants until the recruitment and familiarisation stages are finished.

## Pre-experimental monitoring

To standardise experiments, participants will be asked to keep their physical activity levels on pre-test days consistent, and to record details about their physical activity, diet, and sleep in a standardised diary. They will also be asked to wear accelerometers that monitor their physical activity and sedentary time during the day (hip-worn Actigraph GT3X+ or wGT3X-BT) and sleep at night (wrist-worn Actigraph GT3X+ or wGT3X-BT). Variables from Actigraph will include total time spent in physical activity and percentages of time in sedentary, light, and moderate-to-vigorous physical activity, according to the standardised Evenson cut-points [46]. Throughout the night on pre-test nights, participants will wear the Actigraph GT3X+ on their left wrist [47]. An algorithm for sleep-wake scoring in children will be used [48], and variables of sleep analysed will include: time to sleep onset, sleep efficiency, times waking up after sleep onset, and total sleep. The diaries will provide complementary information regarding times of waking up, going to bed, and sleep quality. Measured physical activity, sedentary time, and sleep parameters will be used to confirm compliance to the study protocol and for adjustment in statistical analyses.

Within the diary, dietary information will be recorded by the participants to standardise dietary intake on pre-test days. A diet and nutrition analysis software (Dietist Net Pro, http://www.kostdata.se/se/dietist-net/dietist-net-pro) will be used to analyse the nutrient intake of pre-test day meals of the participants. Participants will be asked to avoid foods that are rich in nitrate on pre-test days and fast after dinner until they come to GIH on test mornings. Additionally, participants will be asked to not consume chewing gum or use anti-bacterial mouthwash on test days as these can inhibit oral nitrate metabolism [4, 49]. One of the researchers will send a text message to the participants two days before the test days to remind them about the standardisations mentioned.

## Experimental conditions

Upon arrival to the laboratory, pre-experimental monitoring measures will be collected (e.g. physical activity and diet diary) and height measured. Participants will arrive at GIH in a taxi (organised by GIH) to be consistent with energy expenditure and physical activity behaviour. Weight will be measured on the day of the high-nitrate breakfast condition. This day was chosen since nitrate provided will be individualised based on body weight. The participants will perform a practice test of the working memory cognitive task, provide a saliva sample (to assess nitrate/nitrite levels), and be asked to answer questions related to psychological factors (mood, alertness, and sleepiness) before the experimental breakfast conditions (**Fig 2**). The three experimental conditions of this study are as follows:

A. No breakfast

B. Low-nitrate breakfast (regular breakfast)

C. High-nitrate breakfast (regular breakfast supplemented with nitrate provided through concentrated beetroot juice)

On the first breakfast test-day (either condition B or C), participants will be able to choose how much food they want to consume from the foods provided. Foods will include: wholemeal bread, cheese, ham, cucumber, yoghurt, and muesli with low sugar content. On subsequent

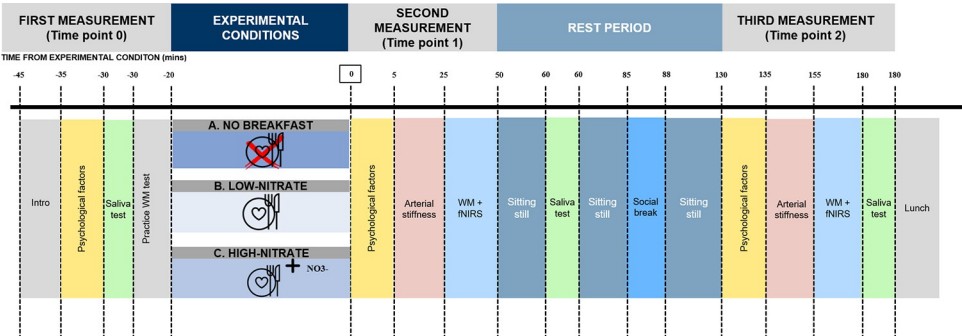

**Fig 2. Timeline of measurements and experimental conditions.** WM = working memory; fNIRS = functional near-infrared spectroscopy.

visits the same and exact amount of food (breakfast will be weighed and documented) will be given to the participants. The breakfast consumption should be as close to the participant's habitual diet as possible, hence why participants can choose the amount of breakfast that they deem will satisfy them. The breakfast provided will be the same in both breakfast conditions and will not contain foods that have a high glycaemic index (e.g. white bread), to reduce the effects of glucose. Previous research has indicated that postprandial blood glucose concentrations could mediate the effects of breakfast on cognitive function [33]. However, a small amount of orange juice will be provided in both breakfast conditions; primarily to help wash down the taste of the concentrated beetroot juice in the high-nitrate breakfast condition. Participants are allowed to drink water ad libitum throughout the course of each experimental day; however, amounts of fluid will be attempted to be matched for each experimental condition. The amount of nitrate provided in the high-nitrate breakfast condition will be adjusted for weight of the participant and they will be asked to drink all the concentrated beetroot juice. Each 60 ml concentrated beetroot juice shot (product of Umara AB, Sweden) contains 500 mg of nitrate. The amount of nitrate provided will be based on the daily acceptable limit of 3.7 mg per kg body weight per day [50], so for a 40 kg individual the amount of nitrate would be 148 mg. In the scenario that the participant does not consume all the beetroot juice, then the amount they consumed will be documented. There should be very little variation in nitrate content between the shots as they are designed to contain 500 mg of nitrate per 60 ml of concentrated beetroot juice and are all taken from the same batch.

There will be an 80-minute sitting period between the first and second measurements of working memory, cognitive task-related CBF, and arterial stiffness (**Fig 2**). In other words, 130 min from breakfast to the last measurement. The period was determined based on previous research in adults showing that plasma nitrite from beetroot juice peaks at 2.5–3 hrs after ingestion [5]. During the sitting period participants will be able to read books or listen to audio books that can be related to their schoolwork but will not be allowed to use any technological devices nor partake in any physical activity that could lead to arousal. One social break will be provided on each test day, at the same time each test day, whereby the participant can talk to one of the researchers for a duration of 2–3 minutes, which will be consistent for each participant.

## Outcome measures

**Primary outcome:** Working memory.

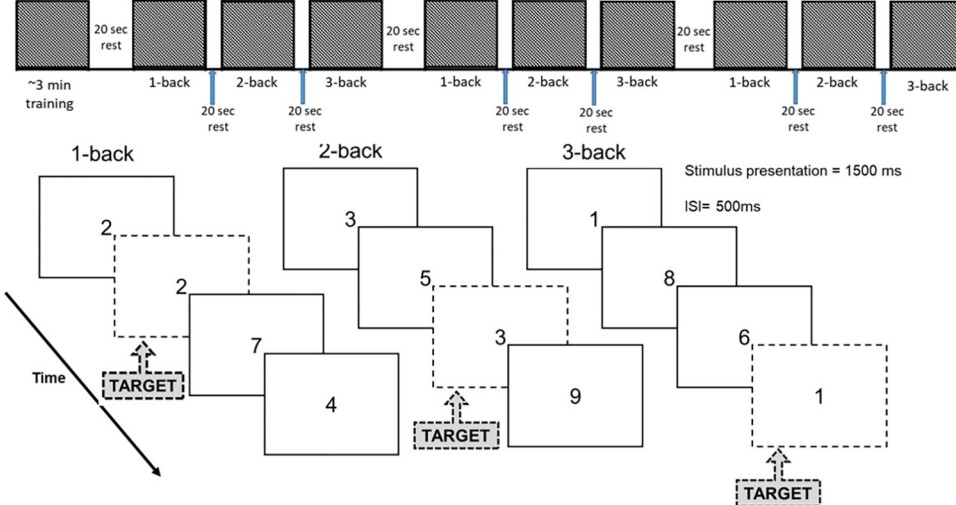

**Fig 3. Schematic of the working memory tasks to measure cognitive function.** Adapted from Heiland et al., 2022 [52]. ISI = inter-stimulus interval.

**Secondary outcomes:** Cognitive task-related CBF, arterial stiffness, psychological factors (mood, alertness, and sleepiness), and salivary nitrate/nitrite concentrations.

**Working memory.** N-back paradigms will be used to assess working memory performance, consisting of the 1-, 2-, and 3-back tests [51] at two time points (time points 1 and 2) after the breakfast experimental conditions (**Fig 2**). The n-back paradigms are computerised numerical versions where the participants will indicate if the digit presented on the screen is the same digit that was presented 1 digit earlier (1-back), 2 digits earlier (2-back), or 3 digits earlier (3-back) (**Fig 3**). Participants will indicate this via a key press within a 2000 ms period from the onset of the digit, whereby each digit is presented for 1500 ms with an inter-stimuli interval of 500 ms. E-Prime 2.0 (Psychology Software Tools) was used to create the n-back tests and for capturing the data. Before each n-back test, participants will be required to stare at a white dot on a black screen and count from zero to reduce mind-wandering for 20 secs, which will be used as the baseline. Average reaction time (m/s) of correct responses and accuracy (number of correct responses) will be measured for each n-back test across the 3 blocks of 20-digit sequences each, with each n-back test lasting 35 secs.

**Cerebral blood flow (CBF).** Cognitive task-related CBF will be measured as changes in oxy-Hb and deoxy-Hb via near-infrared light using a non-invasive, multi-channel, continuous wave fNIRS instrument (portable NIRSport, 8–8 system, with short-separation channels, NIRx Medizintechnik GmbH, Berlin, Germany). CBF parameters will be measured at two time points (time points 1 and 2) during the working memory tasks (**Fig 2**). The optodes for this instrument will be placed over the prefrontal cortex and data will be sampled at 7.81 Hz at wavelengths 760 nm and 850 nm. The fNIRS cap has 16 channels created by 8 LED light sources and 8 detectors that are placed according to the standard 10–20 system, with a source-detector separation distance of 3 cm for long-separation channels. Additional short-separation channels with a distance of 0.8 cm, to account for physiological confounders [53], will also be used. CBF was chosen to be measured in the prefrontal cortex because working memory tests such as the n-back tasks have been shown in previous studies to mainly involve this region [53–55]. NIRStar 15.2 software will be used with a predefined montage (Heiland et al. [52] Additional file 2) to record the CBF data during the working memory tasks.

After placing the fNIRS cap on the participant's head, its symmetrical placement will be ascertained by visually ensuring that the Cz point is centrally located and that the distance between the nasion and the Fpz point are aligned. The cap will then be snuggly closed using the chinstrap. System calibration will be performed prior to the cognitive task, after dimming the test room lights to reduce the effects of ambient light, to optimise signal quality. If signal quality is low, then the positioning of the optodes that form the faulty channels will be corrected by moving hair with a cotton swab and/or gel, or the cap will be totally removed and replaced, with subsequent recalibration. After calibration, at time point 1, a mark will be placed on the forehead of the participant just below the cap edge to ensure replicability in the placement of the cap for time point 2. To minimise the auditory distractions while performing the cognitive task, headphones will be recommended to the participant at the initial measurement and subsequently used or not in accordance with the first measurement occasion for consistency. FNIRS signals will be visually inspected for quality, lost channels, and motion artefacts during measurements and any divergences noted.

**Psychological factors.** Psychological factors will be assessed while participants are relaxed and seated and will occur before the experimental conditions (**Fig 2. Time point 0**) and at two subsequent time points (**Fig 2. Time points 1 and 2**). Participants' mood will be recorded using the Positive and Negative Affect Scale (PANAS) [56]. PANAS has been validated in adolescents and is shown to be a reliable measure of moods [57], with 10 positive and 10 negative moods. Participants will rate their mood on a 5-point scale (very slightly/not at all; a little; moderately; quite a bit; extremely) and will represent how they feel at that moment. The positive affects include: interested, excited, strong, enthusiastic, proud, alert, inspired, determined, attentive, and active; while the negative affects include: distressed, upset, hostile, irritable, scared, jittery, afraid, ashamed, guilty, and nervous. Scores for the positive affects will be added up; providing one overall value and ranging from 10–50 with higher scores indicating higher positive mood. Scores for the negative affects will be added up; providing one overall score and ranging from 10–50 with higher scores indicating more negative mood [56]. Alertness will be measured using a 10 cm visual analogue scale (VAS); ranging from "not at all" to "completely alert" [58]. VAS has previously shown high reliability and validity among adults [59], and also among adolescents [60, 61]. A highly valid measure of sleepiness, the Karolinska Sleepiness Scale questionnaire (KSS) [62, 63], will be used to assess levels of sleepiness via a 9-point Likert scale, ranging from "extremely alert" to "very sleepy, great effort to keep awake, fighting to sleep"; with higher scores indicating higher levels of sleepiness.

**Arterial stiffness.** Arterial stiffness will be measured via two variables: pulse wave velocity (PWV) and augmentation index (AIx) using SphygmoCor XCEL PWA/PWV system [64] at two time points (time points 1 and 2) after the experimental conditions (**Fig 2**). The SphygmoCor XCEL PWA/PWV system has been tested for validity in this age group [65]. While AIx is an indirect measure of arterial stiffness, PWV is considered to be the gold standard [52]. AIx is defined as the difference between the first and second systolic peak and is expressed as a percentage of the pulse pressure. A similar procedure was performed in an earlier study of adolescents [52]. After 2 min of rest, pulse wave analysis (PWA) will be estimated from a brachial arm cuff where blood pressure and waveforms will be derived. This will be followed by measuring PWV from three high fidelity pressure waveforms using a carotid tonometer with a right leg cuff. PWV will be assessed between carotid and femoral arteries and is calculated by the formula: PWV (m/s) = distance between measurement location (m) / transit time (s) [30]. The average of the three recordings will be used to determine PWV (m/s) [66].

**Saliva nitrate/nitrite concentrations.** Nitrate and nitrite concentrations will be collected via three saliva samples: one before the experimental conditions (time point 0) and at two time points (time points 1 and 2) after the experimental conditions (**Fig 2**). Cotton buds in test

tubes will be used to collect the saliva. These samples will be centrifuged and stored at -80˚C, until the time of analysis (<6 months from collection). The OxiSelect™ In Vitro Nitric Oxide (Nitrite/Nitrate) Assay Kit (Fluorometric) from Cell Biolabs, Inc. (San Diego, CA, USA) will be used to measure nitrate ($NO_3^-$) and nitrite ($NO_2^-$) levels collected from the saliva samples.

## Data management

All participants will be assigned to a unique identifier number to remain pseudonymous. All data files with personal data will be stored in an encrypted file format and only researchers will have access to the key-coded file. Original paper files will be stored on site and locked in cabinets. Data will be entered electronically on-site in a password-protected file and data will be double checked for errors, with changes being documented. Consistency checks will be carried out for data integrity.

## Sample size

To the best of our knowledge no studies have investigated the acute effects of nitrate consumption on cognitive function nor vascular health in adolescents. Therefore, an adult population was used to calculate effect sizes for the effect of nitrate on these outcomes, however, the adolescent population was used to calculate effect sizes for the effect of breakfast on working memory and psychological outcomes. Arterial stiffness and CBF mechanistic outcomes have less variance than cognitive performance and psychological factors, thus the sample size was calculated from effect sizes of working memory performance and psychological outcomes. Effect sizes were calculated on G*Power software (Franz Faul, Universität Kiel, Germany, v 3.1.9.2) using differences between two dependent means and standard deviations. The sample size was calculated: using $\alpha = 0.05$, $\beta = 0.8$, correlation = 0.5 and assuming a two-tailed test. The effect size based on the effects of nitrate on working memory = 0.63, giving a sample size = 22. The effect size based on the effects of breakfast on psychological factors ranged from 0.67 to 0.48, giving a sample size range between 20 and 36. Our sample size will thus be the largest number found here with seven additional participants to account for 20% drop out rate, thus n = 43.

## Statistical methods

**Working memory, psychological factors, arterial stiffness, and nitrate/nitrite analysis.** Linear mixed effect models, with an unstructured covariance structure, will be used to estimate within condition changes in working memory reaction time and accuracy, psychological factors, arterial stiffness, and nitrate/nitrite concentrations from time point 1 to time point 2 using Stata/SE 17.0 (StataCorp, LLC, Texas, USA). Additional models will be performed with condition and time interactions to assess between condition changes in mean differences in these outcomes.

**Cerebral blood flow analysis.** The task-related relative changes (Δμmol) in oxy-Hb and deoxy-Hb will be estimated using MATLAB-based software NIRS Brain AnalyzIR Toolbox (https://github.com/huppertt/nirs-toolbox [67]). Oxy-Hb will be the predominant cerebral blood flow outcome as it has previously been reported to be the most sensitive indicator of neural activation [68]. FNIRS probe registration will be individualised to account for anatomical differences and probe placement, by using head circumference, nasion-inion, and pre-auricular distances from each participant. Quality check of each signal will be initially performed, including checking the relative coefficient of variation (CV%) for each channel. A CV% less than 15% will indicate good signal quality. Poor signal quality will be dealt with in the statistical analysis. Raw voltage will then be converted to optical density and subsequently to haemoglobin concentrations based on the modified Beer-Lambert law [52]. Data will be

processed and analysed in a similar manner outlined in Heiland et al. [52]. Minimal manipulation will be performed on the signals as suggested by Santosa et al. [67]. First level statistics will involve estimating the fNIRS parameters during the 1-, 2-, and 3-back tasks (each 35 sec duration) relative to the baseline (20 sec rest). Brain activation will be predicted using a general linear model (GLM) with a canonical design matrix and an autoregressive pre-whitening approach with iteratively reweighted least squares (AR-IRLS) for each source-detector pair. The AR-IRLS approach will be used because it reduces the false-discovery rate [69]. Short-separation regressors will be used to control for type I errors caused by systemic physiology and motion artefacts. The GLM will yield regression coefficients (betas) for each channel in each n-back task and condition that will be used in the second-level analyses (group-level analysis). This will entail the coefficients of Δoxy- and Δdeoxy-Hb each being averaged across the prefrontal region, and separately by left and right prefrontal hemispheres for each n-back task and condition. A False Discovery Rate (FDR) correction using a Benjamini-Hochberg procedure will be used to correct for multiple comparisons (*q-value* = FDR-adjusted $p \leq 0.05$). Linear mixed effect models will be employed with Δoxy- and Δdeoxy-Hb as dependent variables, with condition and time as fixed effects and subject as a random effect to assess within and between condition differences in CBF parameters comparing the n-back tasks. Linear mixed effects models with maximum likelihood estimation will help to deal with potential missing data. Number of missing data will also be reported and checked for missing data mechanisms. Additional, linear mixed models will be performed to assess intervention effects including time and condition as interaction terms.

## Discussion

This study will provide understanding into whether increased nitrate intake can improve working memory performance in adolescents. Furthermore, insight into changes in task-related CBF as a possible mechanism explaining the effects of nitrate on cognitive performance will be explored and if improvements will also be observed in arterial stiffness and psychological well-being. Previous studies have only investigated these acute effects in adults, resulting in mixed results [14]. This will be the first study to be performed in adolescents.

In addition, the effect of having breakfast or skipping breakfast and whether this mealtime should be promoted for adolescent's cognitive performance, vascular, and psychological health will be investigated. Previous acute studies in adolescents have shown mixed findings on the effects of breakfast compared to no breakfast on memory, attention, and executive function [33]. Many of these studies have not reported important factors, such as how much breakfast was consumed, fasting requirements, as well as not controlling for food consumed the evening before [33]. Thus, it is important to further investigate the effect of breakfast on working memory and control for these methodological factors to provide more robust findings. It is also crucial to investigate the acute effects of breakfast on CVD-related risk factors using intervention studies. Previous research has found regular breakfast consumption to have positive benefits on CVD-related risk factors [35, 36], which is promising, however not causal. If positive acute effects of breakfast on CVD-related risk factors can be shown in this study, then this could pave the way for intervention studies of longer duration that focus on the effect of habitual breakfast consumption and risk factors related to CVD. This would provide more evidence into the relationship between regular breakfast consumption and CVD-related risk factors.

There are many strengths of this study. First is the crossover, randomised study design will provide results of high statistical power and will lower the risk of confounding errors since all participants will be their own controls. The use of modern technology in measuring task-related CBF by fNIRS with short separation channels with advanced statistical approaches,

which is recommended for optimal results [70], is another strength of this study. This technology offers non-invasive quantification of cortical haemodynamic responses [71] and has a higher degree of tolerance to motion artefacts in comparison to other neuro-imaging instruments [72]. The SphygmoCor XCEL system for measuring PWV and AIx has been validated and shown to provide accurate results [73] and follows the guidelines laid out by the Artery Society Guidelines [74]. The standardisation of physical activity and diet via a diary and objectively measuring physical activity are additional advantages of this study.

Participants may not adhere to drinking beetroot juice, which is a. However, providing regular breakfast foods to be consumed with the nitrate-rich beetroot juice and providing a small glass of orange juice to wash away the taste will help to resolve this issue. Participants may also become distracted during the cognitive task, nonetheless headphones to protect from surrounding noise will be provided, tests will be conducted in a non-disruptive laboratory setting, and mind-wandering will be reduced with counting tasks. The lack of pseudo-randomisation of n-back tasks of different loads may also lead to some anticipatory effects, however, the block-design and randomised crossover nature of this study may help minimise this. In addition, maintaining an order of the task loads can help reduce confusion during the testing. Results may also be affected by the participants socialising in between tests, thus they will be continuously monitored.

This study will be informative on the acute effects of nitrate as well as breakfast on working memory, cardiovascular risk factors, and psychological well-being. Consequently, this may have important implications for school academic achievement. Furthermore, the results could influence dietary recommendations on nitrate-rich foods and breakfast consumption in this population. It is vital to investigate these effects early in life as improvements in health could translate into adulthood. This may help to reduce the risk of future cognitive impairment and cardiovascular health problems; thus, having significant implications on public health. It will also be a starting point for more research looking at the effects of breakfast and its composition (e.g. nitrate-rich) on cognitive and vascular outcomes. In turn, results can inform public health interventions and help in the development of recommendations to improve the health of adolescents.

## Trial status

This is the first version of the main study protocol. Recruitment commenced in January/February 2022, and data collection commenced in late February 2022.

## Supporting information

**S1 Checklist. SPIRIT 2013 checklist: Recommended items to address in a clinical trial protocol and related documents.**
(PDF)

## Author Contributions

**Conceptualization:** Callum Regan, Olga Tarassova, Karin Kjellenberg, Filip J. Larsen, Hedda Walltott, Maria Fernström, Gisela Nyberg, Maria M. Ekblom, Björg Helgadóttir.

**Funding acquisition:** Örjan Ekblom, Gisela Nyberg.

**Methodology:** Callum Regan, Emerald G. Heiland, Örjan Ekblom, Olga Tarassova.

**Project administration:** Örjan Ekblom, Maria M. Ekblom, Björg Helgadóttir.

**Resources:** Örjan Ekblom, Maria Fernström, Maria M. Ekblom.

**Software:** Olga Tarassova.

**Supervision:** Emerald G. Heiland, Maria M. Ekblom, Björg Helgadóttir.

**Writing – original draft:** Callum Regan, Emerald G. Heiland, Örjan Ekblom, Olga Tarassova, Karin Kjellenberg, Filip J. Larsen, Hedda Walltott, Maria Fernström, Gisela Nyberg, Maria M. Ekblom, Björg Helgadóttir.

**Writing – review & editing:** Callum Regan, Emerald G. Heiland, Örjan Ekblom, Olga Tarassova, Karin Kjellenberg, Filip J. Larsen, Hedda Walltott, Maria Fernström, Gisela Nyberg, Maria M. Ekblom, Björg Helgadóttir.

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
