## [Decision Letter · Decision Letter 0]

14 Feb 2023

PONE-D-22-31135

Acute effects of nitrate and breakfast on working memory, cerebral blood flow, arterial stiffness, and psychological factors in adolescents: Study protocol for a randomised crossover trial

PLOS ONE

Dear Dr. Heiland,

Thank you for submitting your manuscript to PLOS ONE. After careful consideration, we feel that it has merit but does not fully meet PLOS ONE’s publication criteria as it currently stands. Therefore, we invite you to submit a revised version of the manuscript that addresses the points raised during the review process.

This decision was based partly on my independent review and additional reviewers. Please note that I have acted as a reviewer for this manuscript, and you will find my comments below, under Reviewer 3

We look forward to receiving your revised manuscript.

Kind regards,

Shashank Shekhar

Academic Editor

PLOS ONE

Journal Requirements:

2. Please note that in order to use the direct billing option the corresponding author must be affiliated with the chosen institute. Please either amend your manuscript to change the affiliation or corresponding author, or email us at plosone@plos.org with a request to remove this option.

Reviewers' comments:

Reviewer's Responses to Questions

**Comments to the Author**

1. Does the manuscript provide a valid rationale for the proposed study, with clearly identified and justified research questions?

Reviewer #1: Yes

Reviewer #2: Yes

Reviewer #3: Yes

2. Is the protocol technically sound and planned in a manner that will lead to a meaningful outcome and allow testing the stated hypotheses?

Reviewer #1: Yes

Reviewer #2: Yes

Reviewer #3: Partly

3. Is the methodology feasible and described in sufficient detail to allow the work to be replicable?

Reviewer #1: Yes

Reviewer #2: Yes

Reviewer #3: Yes

4. Have the authors described where all data underlying the findings will be made available when the study is complete?

Reviewer #1: Yes

Reviewer #2: No

Reviewer #3: Yes

5. Is the manuscript presented in an intelligible fashion and written in standard English?

Reviewer #1: Yes

Reviewer #2: Yes

Reviewer #3: Yes

6. Review Comments to the Author

You may also provide optional suggestions and comments to authors that they might find helpful in planning their study.

Reviewer #1: Interesting topic to study. The author provide detail information on what will they do for the study. Recommended to inform the reliability of the instruments that will be used especially in measuring the psychology part.

Reviewer #2: The aim of this randomized crossover study using three conditions is to investigate acute effects of nitrate and breakfast on working memory, task-related cerebral blood flow, arterial stiffness, and psychological outcomes in adolescents. The outcomes will be repeatedly measured.

Minor revisions:

1- State the software or system in which the electronic data will be captured.

2- Line 495: Indicate the underlying covariance structure that will be used in the linear mixed effect models or criteria that will be used to choose the structure.

Reviewer #3: This is a randomized crossover clinical trial design protocol to explore the effect of nitrates in the breakfast on working memory as well cerebral blood flow, psychological factors and arterial stiffness.

Its a good reading going over the study design. Mostly the study design is rigorous except for the fNIRS design. I see two issues with this design as below and modifying it may increase the overall quality of data.

1. Currently the rest period is not randomized. So randomizing the duration of rest periods will be more optimized.

2. randomizing the order of different load conditions in a counterbalance order will help get a more robust data.

7. PLOS authors have the option to publish the peer review history of their article (what does this mean?). If published, this will include your full peer review and any attached files.

Reviewer #1: No

Reviewer #2: No

Reviewer #3: No

---

## [Author Response · Author response to Decision Letter 0]

9 Mar 2023

Reviewer's Responses to Questions

Have the authors described where all data underlying the findings will be made available when the study is complete?

RESPONSE

The data from this study are restricted. These restrictions are mentioned in the Methods section under “Availability of data and materials”, stating that the data can be made available upon reasonable request by contacting the principal investigator of the study, Björg Helgadóttir. This can be found in the manuscript (lines 548-550 in the track changed manuscript).

Review Comments to the Author

Reviewer #1: Interesting topic to study. The author provide detail information on what will they do for the study. Recommended to inform the reliability of the instruments that will be used especially in measuring the psychology part.

RESPONSE

Thank you for your comment. The PANAS, used to measure mood, has been tested for reliability and validity as stated in lines 437-438 and reference 58. We have now added in some references for the Karolinska Sleepiness Scale questionnaire (line 449; reference 64) and for VAS (lines 448-449; references 60-62).

Reviewer #2: The aim of this randomized crossover study using three conditions is to investigate acute effects of nitrate and breakfast on working memory, task-related cerebral blood flow, arterial stiffness, and psychological outcomes in adolescents. The outcomes will be repeatedly measured.

Minor revisions:

1- State the software or system in which the electronic data will be captured.

2- Line 495: Indicate the underlying covariance structure that will be used in the linear mixed effect models or criteria that will be used to choose the structure.

RESPONSE

1. Thank you for your comments. The softwares and systems are mentioned throughout the text:

a. Cognitive tests: E-prime 2.0 to create and capture (lines 392-393); analyzed with Stata (line 511).

b. CBF: captured via NIRStar 15.2 (line 416) and analyzed with MATLAB-based software NIRS Brain AnalyzIR Toolbox (line 517).

c. Arterial stiffness: SphygmoCor XCEL PWA/PWV system (line 457); and analyzed with Stata (line 511)

2. The linear mixed models will be run with an unstructured covariance structure, as it is more flexible, not putting any constraints on the model. This has been now included in the statistical analysis section (line 508). 

Reviewer #3: This is a randomized crossover clinical trial design protocol to explore the effect of nitrates in the breakfast on working memory as well cerebral blood flow, psychological factors and arterial stiffness.

Its a good reading going over the study design. Mostly the study design is rigorous except for the fNIRS design. I see two issues with this design as below and modifying it may increase the overall quality of data.

1. Currently the rest period is not randomized. So randomizing the duration of rest periods will be more optimized.

2. randomizing the order of different load conditions in a counterbalance order will help get a more robust data.

RESPONSE

1. Thank you for your comments. Yes, it has been previously suggested in the literature that varying the length of rest periods can help to reduce resonance effects. Moreover, in fNIRS studies of block-design, it has been recommended that the baseline rest periods, when the stimulus-evoked cortical hemodynamic response goes back to resting levels, should be approximately the same duration as the stimulus (Herold et al. 2018 [Reference 71]). Therefore, we have a consistent duration for our baseline rest periods, and believe it is adequate for the hemodynamic response to return to baseline levels. Unfortunately, at this point we cannot make any changes to the tests, but we may include this in the discussion of our results later.

2. We agree that the lack of randomization of the different loads of the n-back test may be considered a limitation, and we have discussed this in our group when designing the study. However, due to the three load conditions of the n-back, randomization will increase confusion on what test the participants are performing during testing, thus, we decided to keep the order and that the block-design and the randomized crossover design will help to minimize any anticipatory effects that could occur from the lack of randomization of the n-back tests. I have now added this in the manuscript (lines 594-597) and this will also be considered in the discussion around the results later.

---

## [Decision Letter · Decision Letter 1]

27 Apr 2023

Acute effects of nitrate and breakfast on working memory, cerebral blood flow, arterial stiffness, and psychological factors in adolescents: Study protocol for a randomised crossover trial

PONE-D-22-31135R1

Dear Dr. Heiland,

We’re pleased to inform you that your manuscript has been judged scientifically suitable for publication and will be formally accepted for publication once it meets all outstanding technical requirements.

Kind regards,

Shashank Shekhar

Academic Editor

PLOS ONE

Additional Editor Comments (optional):

I was one of the reviewer during first round and suggested minor revision. I accept your reply to my comments.

Please proof read for some minor errors in the text.

Reviewers' comments:

Reviewer's Responses to Questions

**Comments to the Author**

1. Does the manuscript provide a valid rationale for the proposed study, with clearly identified and justified research questions?

Reviewer #2: Yes

2. Is the protocol technically sound and planned in a manner that will lead to a meaningful outcome and allow testing the stated hypotheses?

Reviewer #2: Yes

3. Is the methodology feasible and described in sufficient detail to allow the work to be replicable?

Reviewer #2: Yes

4. Have the authors described where all data underlying the findings will be made available when the study is complete?

Reviewer #2: No

5. Is the manuscript presented in an intelligible fashion and written in standard English?

Reviewer #2: Yes

6. Review Comments to the Author

You may also provide optional suggestions and comments to authors that they might find helpful in planning their study.

Reviewer #2: All comments have been adequately addressed.

7. PLOS authors have the option to publish the peer review history of their article (what does this mean?). If published, this will include your full peer review and any attached files.

Reviewer #2: No

---

## [Editor Report · Acceptance letter]

12 May 2023

PONE-D-22-31135R1 

Acute effects of nitrate and breakfast on working memory, cerebral blood flow, arterial stiffness, and psychological factors in adolescents: Study protocol for a randomised crossover trial 

Dear Dr. Heiland:

I'm pleased to inform you that your manuscript has been deemed suitable for publication in PLOS ONE. Congratulations! Your manuscript is now with our production department. 

Kind regards, 

on behalf of

Dr. Shashank Shekhar 

Academic Editor

PLOS ONE